# Insights into Characteristic Volatiles in Wuyi Rock Teas with Different Cultivars by Chemometrics and Gas Chromatography Olfactometry/Mass Spectrometry

**DOI:** 10.3390/foods11244109

**Published:** 2022-12-19

**Authors:** Yue Zhang, Suyoung Kang, Han Yan, Dongchao Xie, Qincao Chen, Haipeng Lv, Zhi Lin, Yin Zhu

**Affiliations:** 1Tea Research Institute, Chinese Academy of Agricultural Sciences, Hangzhou 310008, China; 2Key Laboratory of Tea Processing Engineering of Zhejiang Province, Hangzhou 310008, China

**Keywords:** characteristic volatiles, cultivars, aroma-active compounds, GC-O/MS, multivariate statistical analysis, Wuyi rock teas

## Abstract

Wuyi rock tea (WRT) is one of the most famous subcategories of oolong tea, exhibiting distinct aroma characteristics with the application of different cultivars. However, a comprehensive comparison of the characteristic volatiles among WRTs with different cultivars has rarely been carried out. In this study, non-targeted analyses of volatile fragrant compounds (VFCs) and targeted aroma-active compounds in WRTs from four different cultivars were performed using chemometrics and gas chromatography olfactometry/mass spectrometry (GC-O/MS). A total of 166, 169, 166, and 169 VFCs were identified for Dahongpao (DHP), Rougui (RG), Shuixian (SX), and Jinfo (JF), respectively; and 40 components were considered as the key differential VFCs among WRTs by multivariate statistical analysis. Furthermore, 56 aroma-active compounds were recognized with predominant performances in “floral & fruity”, “green & fresh”, “roasted and caramel”, “sweet”, and “herbal” attributes. The comprehensive analysis of the chemometrics and GC-O/MS results indicated that methyl salicylate, *p*-cymene, 2,5-dimethylpyrazine, and 1-furfurylpyrrole in DHP; phenylethyl alcohol, phenethyl acetate, indole, and (*E*)-β-famesene in RG; linalool, phenethyl butyrate, hexyl hexanoate, and dihydroactinidiolide in JF; and naphthalene in SX were the characteristic volatiles for each type of WRT. The obtained results provide a fundamental basis for distinguishing tea cultivars, recombination, and simulation of the WRT aroma.

## 1. Introduction

Wuyi rock tea (WRT) is a representative type of northern Fujian oolong tea of excellent quality, which refers to the oolong teas produced in the administrative area of Wuyishan City (Fujian Province, China). Owing to its unique floral aroma, mellow sweet taste and deep cultural connotation, WRT is recognized as one of the most famous and top-ranking subcategories of oolong tea, which has become a favorite non-alcoholic drink in China and overseas [1]. Usually, commercial Wuyi rock teas are categorized as “Rougui” (RG), “Shuixian” (SX), “Dahongpao” (DHP), “Jinfo” (JF), “Tieluohan”, etc. according to their cultivars [2], all of which present a similar “rock flavor” as the primary impression. Nowadays, “rock flavor” is recognized as a proper noun of the excellent flavor quality of Wuyi rock tea [3]. The common “rock flavor” of WRTs was considered to be formed during tea processing especially after the full firing process, and *N*-containing volatiles, e.g., 2-ethyl-3,5-dimethylpyrazine, were identified as the corresponding aroma-active compounds [3,4]. The variation in aroma profiles and aroma-active compounds during WRT processing has been fully investigated in RG [5,6,7,8], SX [3,9], and other cultivars [10]. Conversely, the distinctive flavor of each type of WRT mainly originates from the material basis of its own cultivars [1]. However, the aroma of Wuyi rock teas is difficult to distinguish for ordinary consumers, while for professional sensory evaluation assessors, different characteristic scents associated with their own cultivars can be recognized, e.g., orchid-like (SX), cinnamon-like (RG), floral (DHP), and fruity (JF), etc. [11]. Similarly, the appearance of the shape and the taste of WRTs are also easily confused, which might bring some opportunistic business for lawless merchants. For the purpose of the scientific understanding of different WRTs and the protection of the corresponding brands, the discrimination of different WRTs through advanced and objective techniques is needed. Therefore, the characterization of aroma components in Wuyi rock teas with different cultivars is meaningful and beneficial for the better understanding and discrimination of different WRTs. Recently, aroma profiles of oolong tea with different cultivars, e.g., Tieguanyin, Benshan, Maoxie, Huangjingui, and Jinguanyin [12]; Huangzhixiang and Huangdan [13]; and SX, Huangmeigui, and Zimudan [14] have been studied and distinguished, most of which were grown in southern Fujian and Guangdong province of China. In addition, a comparison of the aroma profiles of WRTs in the same growing regions (northern Fujian) with similar cultivars has rarely been reported. Moreover, the aroma-active compounds in the different WRTs are mainly determined by the assessment of the odor activity value (OAV) [14,15], while the direct contribution evaluation of each volatile in WRTs with similar cultivars by gas chromatography olfactometry (GC-O) is rarely performed, which has been regarded as an intuitive and essential technology in the flavor analysis of foods [16,17,18].

Therefore, the aims of this study were to verify the key differential VFCs and characteristic odorants in WRTs with four representative cultivars by headspace solid-phase microextraction (HS-SPME)-GC-O/MS, chemometrics, and multivariate statistical analysis to comprehensively uncover the aroma essence of each WRT and provide a scientific reference for the discrimination of tea cultivars.

## 2. Materials and Methods

### 2.1. Reagents and Materials

Chemical standards hexanal, *N*-ethylpyrrole, 2-methylpyrazine, heptanal, 2,5-dimethylpyrazine, 2-acetylfuran, dimethyl trisulfide, 1-octen-3-ol, methylheptenone, octanal, (*E*,*E*)-2,4-heptadienal, 2,2,6-trimethyl-cyclohexanone, benzeneacetaldehyde, (*E*)-2-octenal, 2-acetylpyrrole, 3,5-octadien-2-one, methyl benzoate, linalool, nonanal, 6-methyl-3,5-heptadiene-2-one, phenylethyl alcohol, (*E*)-2-nonenal, benzyl acetate, methyl salicylate, (*E*,*E*)-2,4-nonadienal, indole, hexyl hexanoate, (*E*)-2-hexenyl hexanoate, α-cedrene, α-ionone, coumarin, (*E*)-β-famesene, *trans*-β-ionone, dihydroactinidiolide, limonene, linalool, α-terpineol, (*E*)-nerolidol, and (*Z*)-nerolidol were purchased from J&K Scientific Ltd. (Beijing, China), Sigma-Aldrich Corp. (Shanghai, China), Innochem Science & Technology Co., Ltd. (Beijing, China), and Aladdin Corp. (Shanghai, China). *Trans*-linalool oxide (furanoid), *cis*-linalool oxide (furanoid) and *trans*-β-ocimene were purchased from Toronto Research Chemicals Inc. (Toronto, ON, Canada). Calamenene was purchased from Purechem-Standard Biotechnology Co. Ltd. (Chengdu, China). The standards were diluted with ethanol (PR grade; Beijing, China) for chemical structure identification. *N*-alkanes mixed standards (C8–C40) for linear retention index (RI) determination were purchased from J&K Scientific Ltd. (Beijing, China), and purified water, which was used as the tea solvent, was purchased from Wahaha Group Company (Hangzhou, China).

Divinylbenzene/carboxen/polydimethylsiloxane (DVB/CAR/PDMS, 2 cm) purchased from Supelco (Bellefonte, PA, USA) was used as the HS-SPME fiber for extracting volatiles in teas. Threaded headspace bottles (20 mL) were purchased from Agilent Technologies, Inc. (Santa Clara, CA, USA).

### 2.2. Tea Samples

Four representative cultivars of WRT, including DHP, RG, SX, and JF, were used in this study. They were all cultivated in the tea plantations of Yongsheng Tea Industry Co., LTD, near the Wuyi Mountains of Fujian. Among them, DHP, RG, and SX are famous and typical clonal tea cultivars of WRT, while JF belongs to one of the five recognized categories of WRT products named “Qizhong” according to the National Standard “*Product of geographical indication-Wuyi rock-essence tea*” (*GB*/*T 18745-2006*), and it is a mysterious and special cultivar that was carefully bred from sexual cultivars named “Fjjian Caicha”. WRT samples were manufactured in four batches by experienced tea masters according to traditional processing, including withering, making, fixation, rolling, first firing, and full firing. Thereafter, all tea samples were successively crushed, passed through a 40-mesh sieve, and stored at −20 °C for further analysis.

### 2.3. HS-SPME Procedure

Tea powder (1.0 g) and boiling water (5.0 mL) were rapidly placed into a 20 mL threaded headspace bottle. Subsequently, the bottle was automatically placed in a heating oscillator to equilibrate for 3.0 min at 60 °C using an MPS-2 multi-purpose sampler (Gerstel, Mülheim an der Ruhr, Germany) assembled in the GC–MS equipment, and then the DVB/CAR/PDMS fiber was exposed to the headspace of the bottle and stirred at 60 °C for 60 min. Finally, the fiber was desorbed using the GC injector for 5.0 min to release the extracted aroma components. Three replicate experiments were performed for each WRT sample.

### 2.4. GC–MS Analysis

Non-targeted analyses of volatile fragrant compounds (VFCs) in the WRTs were performed using an Agilent 7890B-5977B GC–MS instrument (Agilent, Santa Clara, CA, USA) equipped with an MPS-2 multi-purpose autosampler (Gerstel GmbH & Co. KG, Germany). An HP-5MS column (30 m × 250 μm × 0.25 μm; Agilent, Santa Clara, CA, USA) was used as the gas chromatographic column for the separation of VFCs in different cultivars of WRTs.

The flow rate of the carrier gas (helium with 99.999% purity) was set at 1.6 mL/min. The temperature of the column oven was programmed to increase from 50 °C (held for 2.0 min) to 265 °C (held for 5.0 min) at a rate of 4 °C/min.

MS conditions were as follows: temperature of the transfer line, 250 °C; electron ionization, −70 eV; electron multiplier, 1300 V; ion source temperature, 220 °C; and mass range, *m*/*z* 50–450.

### 2.5. GC-O/MS Analysis

GC-O/MS analysis was performed using an ODP-3 olfactory detection port (Gerstel GmbH & Co. KG, Germany) connected to a GC–MS instrument. The GC and MS parameters for the GC-O/MS analysis were consistent with those for the GC–MS analysis, and the aroma extract was separated equally into the ODP and MS detectors after GC separation. The temperatures of the injector of the ODP and transfer line were set at 150 °C and 200 °C, respectively. High-purity nitrogen (purity ≥ 99.999%) was used as the makeup gas, with a constant flow rate of 50 mL/min.

GC-O analysis of aroma-active compounds in WRTs of four different cultivars was performed by six trained panelists (two males and four females) aged between 24 and 40 years. The skill training of panelists and GC-O analysis were performed based on a slight optimization of those described in our preliminary work [19,20,21]. The panelists were trained for more than 90 h to distinguish different odor characteristics using the following standards: linalool (floral), γ-nonanolactone (sweet), hexanal (green), *cis*-linalool oxide (furanoid) (herbal), 2,6-diethyl-pyrazine (roasted), menthol (minty), and (*Z*)-hex-3-en-1-yl acetate (fruity). The odor characteristics and aroma intensity (AI) values of the separated aroma-active compounds in different WRTs were evaluated by panelists. The AI values were displayed via a handle with a 4-point intensity scale from 1 to 4, which was directly connected to the ODP detector. During the GC-O analysis, the panelists selected different buttons to express the AI values of the separated odorants, where “1” was weak, “2” was moderate, “3” was strong, and “4” was extremely strong [22]. Similarly, odor characteristics with consistent retention times from at least three panelists were accepted and further identified by comparison with the NIST library, retention index, and standards, and their AI values were the average of the corresponding panelists.

### 2.6. Data Processing

The data obtained from GC–MS analysis were preliminarily processed by the built-in MSD Chemstation (Agilent Technologies, Inc., Santa Clara, CA, USA), and structural identification was performed through NIST 2014 library research combined with retention index (RI) validation. The compounds with a similarity >750 as matched from the NIST 2014 library and a difference between reported and calculated RI values of less than 20 were retained and identified as VFCs in WRTs. The quantitative ion peak area of each VFC in each WRT was carefully checked using manual integration.

Significant difference analysis of VFCs between groups was performed using IBM SPSS20.0 (IBM Corp., Armonk, NY, USA). Principal component analysis (PCA), partial least squares discriminant analysis (PLS-DA), and hierarchical cluster analysis (HCA) were carried out using SIMCA-P (Version 14.1, Umetrics, Umea, Sweden) and Multi Experiment Viewer 4.8.1 (Oracle Corp., Redwood Shores, CA). Pie charts, bar graphs and radar graphs were drawn using Excel 2013 (Microsoft Corp., Redmond, WA, USA).

## 3. Results and Discussion

### 3.1. Identification of VFCs in WRTs with Different Cultivars

The VFCs in WRTs with different cultivars were analyzed using HS-SPME/GC–MS, and more than 900 peaks were preliminarily detected for each sample. After identification as described in “*2.6. Data processing*”, 171 VFCs that could be classified into 11 categories according to their chemical structures, including esters, ketones, alkenes, aldehydes, nitrogen compounds, alcohols, aromatic compounds, oxygen heterocyclic compounds, lactones, sulfur compounds, and organic acids, were tentatively identified from the four different cultivars of WRTs, as listed in Appendix A. More specifically, 166, 169, 166, and 169 VFCs were determined for DHP, RG, SX, and JF, respectively. Comparing the previous studies on SX [14] and RG [6] oolong teas, it was found that the number of VFCs was more abundant in our study, which may be due to the differences in extraction methods and instrument precision. However, due to the sole stationary phase of the GC column applied in the study, the identification of VFCs would be further checked using another GC column with a different stationary phase in the further work [23]. In this study, the comparison of retention times and MS data with authentic standards of some important VFCs in WRTs were applied for further identification.

The number and concentration distributions of the different categories of VFCs among the four different cultivars of WRTs were further investigated (Figure 1). Overall, the number distributions of the different chemical categories of VFCs among DHP, RG, SX, and JF were similar, and esters (17%), alkenes (17%), and ketones (16%) presented the largest proportions in all types of WRTs, followed by aldehydes (14–15%), nitrogen compounds (9–10%), alcohols (9%), aromatic compounds (6–7%), and oxygen heterocyclic compounds (5−6%). Conversely, lactones (2%), sulfur compounds (2%), and organic acids (1%) comprised VFCs with minimum number proportions in all types of WRTs. The vast majority of nitrogen compounds belonged to nitrogen heterocyclic compounds (Appendix A), which are sometimes referred to as heterocyclic compounds together with oxygen heterocyclic compounds. Esters, ketones and heterocyclic compounds were reported to be the most numerous compounds in RG WRT by using the consistent HS-SPME/GC–MS approach [6], which is in good agreement with our study. In addition, other research indicated that alcohols and heterocyclic compounds were dominant in SX oolong tea samples [14]. However, in our study, the proportion of alcohols was not as high as reported, possibly because the steam distillation extraction method could extract more low-boiling alcohols, and a larger number of VFCs were identified in this study, resulting in a less prominent proportion of alcohols.

Although it was inappropriate to compare the contents of the VFCs directly using their corresponding peak areas due to the great abundance differences presented in the GC–MS chromatogram brought about by the different functional groups, the peak area of the compounds in the same categories with the same functional groups might provide some references for their concentration distribution orders. Limonene, 1,1,6-trimethyl-1,2-dihydronaphthalene, 2-acetylfuran, dihydroactinidiolide, dimethyl disulfide, and 4-hydroxybutyric acid, which belong to alkenes, aromatic compounds, oxygen heterocyclic compounds, lactones, sulfur compounds, and organic acids, showed the highest concentrations in the corresponding chemical categories in all types of WRTs. Some differences in compounds with the highest concentrations in the other five chemical categories were observed among the four types of WRTs. Methyl salicylate and (*E*,*E*)-3,5-octadien-2-one were the most abundant esters and ketones in DHP, RG, and SX, whereas *cis*-3-hexenyl hexanoate and *trans*-β-ionone replaced them in the corresponding categories in JF. Among aldehydes, furfural showed the highest concentrations in DHP, while (*E*,*E*)-2,4-heptadienal was most abundant in RG, SX, and JF. Among the nitrogen compounds and alcohols, 1-ethyl-1H-pyrrole-2-carboxaldehyde and (*E*)-nerolidol had the highest concentrations in DHP and SX, whereas the concentrations of benzyl nitrile and phenylethyl alcohol were highest in RG and JF.

### 3.2. Screening of Key Differential VFCs among the Four Types of WRTs

To explore the differences in the distribution of VFCs among the four types of WRTs, unsupervised PCA and supervised PLS-DA were applied based on the quantitative ion peak areas of the 171 identified VFCs. As shown in Figure 2A, the WRTs were spontaneously clustered according to their cultivars in the score scatter plot of the PCA model (R^2^X = 0.889, Q^2^ = 0.578). More specifically, the scatters of JF and RG were distributed with slight overlaps, whereas they were clearly distinguished from the DHP and SX samples, which presented a similar trend to the former two types of samples. The above distribution trend might indicate that the chemical basis of the aroma between JF and RG was similar, as well as that between DHP and SX.

Furthermore, the four cultivars of WRTs samples were well distinguished by the PLS-DA model (Figure 2B; R^2^Y = 0.946, Q^2^ = 0.837) using the Pareto scaling model, thereby indicating that some significantly different VFCs existed among DHP, RG, SX, and JF. Moreover, the validation model for the 200 repeated calculations (Figure 2C) showed no overfitting phenomenon in the obtained PLS-DA model (R^2^ = 0.321, Q^2^ = −991). A total of 40 components were further identified as the key differential VFCs among the four types of WRTs, following the selection principle of variable importance in the projection (VIP) values higher than 1.0 in the PLS-DA model [24,25] and *p*-values lower than 0.05, as determined by the Kruskal–Wallis non-parametric test [26]. Thereafter, detailed and visually apparent differences among different groups of WRTs were presented by HCA after normalizing the peak area data. As shown in Figure 3, the concentration distribution of key differential VFCs was visually presented by the heat map, and their distribution trends were generally divided into six classes, including 4, 3, 6, 8, 17 and 2 compounds in classes I–VI.

*trans*-Linalool oxide (furanoid), limonene, *p*-cymene, and *m*-xylene in class I presented higher concentrations in most of the JF and DHP samples, while their concentrations were moderate in the RG samples and extremely low in the SX samples.

Hexyl butyrate, geraniol, and linalool in class II exhibited high concentrations in the JF samples, whereas they were scarce in the other WRTs.

In class III, *cis*-3-hexenyl hexanoate, hexyl hexanoate, (*E*)-2-hexenyl hexanoate, phenethyl butyrate, *cis*-3-hexenyl benzoate, and hexyl benzoate were far more abundant in the JF and RG samples than in the DHP and SX samples, all of which are esters and emit fresh and fruity scents.

In addition to the VFCs in class III, the compounds in class IV, composed of phenylethyl alcohol, (*E*)-β-famesene, (*E*)-nerolidol, phenethyl acetate, benzyl nitrile, phenethyl 2-methylbutyrate, (2-nitroethyl)-benzene, and indole, showed significantly higher concentrations in the RG samples and the lowest concentration levels in the SX samples. However, these compounds were not abundant in the JF samples. Among them, (*E*)-nerolidol and indole are considered as key aroma compounds, which are mainly generated in response to continuous wounding stress during the manufacturing of oolong tea [27,28].

The compounds in the last class (V) showed significantly higher concentrations in the DHP samples than in the other samples, including methyl 2-furoate, *p*-methylacetophenone, methyl salicylate, 1,1,6-trimethyl-1,2-dihydronaphthalene, methyl benzoate, 2-heptanone, mesitylene, 2,5-dimethylpyrazine, (*E*,*E*)-3,5-octadien-2-one, *N*-ethylpyrrole, 1-ethyl-1H-pyrrole-2-carboxaldehyde, 5-methyl furfural, furfural, 1-furfurylpyrrole, benzaldehyde, 2-methylpyrazine, and 2-acetylfuran. The compounds were mainly composed of nitrogen, oxygen heterocyclic compounds and heterocyclic aldehydes, most of which presented a similar nutty and roasted odor. The above heterocyclic compounds have been reported to be correlated with applied heating temperature and time, and they have also been detected in other types of oolong teas [16], and some green [29] and black [30] teas.

Only two compounds, naphthalene and dihydroactinidiolide, were included in class VI. Both of the above compounds showed significantly lower contents in the RG samples than in the other three types of samples; the former compound was more abundant in SX and JF, and the concentrations of the other compound were significantly higher in most of the JF and several of the DHP and SX samples.

In general, it can be summarized that the compounds in classes I and V (DHP), classes I, II, III, and VI (JF), and classes III and IV (RG) were significantly abundant in WRTs, with the corresponding cultivars in parentheses. However, the distributions of the key differential VFCs with higher concentration levels among the SX samples were irregular. Instead, it was consistent that VFCs in classes I–IV were scarce in almost all the SX samples. The clustering analysis of key differential VFCs among the four types of WRTs showed that DHP and SX could be distinguished from JF and RG. This result was consistent with the PCA and PLS-DA trends. Moreover, the above key differential VFCs made different contributions to the tea aroma according to their own flavor characteristics. Some of the key differential VFCs, such as naphthalene (0.44 μg/kg of odor threshold), linalool (6 μg/kg of odor threshold), and methyl salicylate (40 μg/kg of odor threshold), probably had a significant effect on the tea aroma due to their low odor thresholds and high contents [19,22].

### 3.3. Identification of Aroma-Active Compounds in WRTs with Different Cultivars

Although both the aroma profiles and key differential VFCs among WRTs with different cultivars have been identified through stoichiometry, the aroma-active compounds that directly contribute to the aroma quality in each type of WRT are also essential because of the completely different odor thresholds of each VFC. Hence, GC-O/MS analysis of WRTs from four different cultivars was performed. Equal amounts of the four batches of WRTs for each cultivar were mixed and sniffed by six well-trained panelists. As shown in Table 1 and Appendix A, 56 aroma-active compounds were identified, including 54 odorants in the DHP, RG, and SX samples, and 55 odorants in the JF samples. According to the general odor types, the aroma-active compounds could be classified into six classes: 17 odorants in class A with a floral and fruity odor, 16 odorants in class B with green and fresh odor, 9 odorants in class C with roasted and caramel odor, 6 odorants in class D with a sweet odor, and 4 odorants in class E with an herbal scent. In addition, 4 odorants with fatty, sulfurous, or pungent scents belonged to Class F. This classification method of aroma attributes can well compare the odor characteristics between different samples and has often been used in aroma sensory description and aroma recombination and omission experiments [21,22].

Overall, the AIs of the aroma-active compounds ranged from 2.0−3.4 (DHP), 2.3−3.4 (RG), 1.0−3.0 (SX) and 1.5−3.3 (JF) in class A; 2.0−3.3 (DHP), 2.3−3.3 (RG), 1.8−3.3 (SX), and 2.0−3.4 (JF) in class B; 2.3−3.3 (DHP), 2.4−3.0 (RG), 2.4−3.3 (SX), and 2.0−3.2 (JF) in class C; 2.0−3.4 (DHP), 2.4−3.2 (RG), 2.0−3.2 (SX), and 2.3−3.2 (JF) in class D; 2.0−3.0 (DHP), 2.0−2.8 (RG), 2.0−3.0 (SX), and 1.7−3.0 (JF) in class E; 2.0−2.8 (DHP), 2.0−3.3 (RG), 2.3−3.7 (SX), and 2.2−3.0 (JF) in class F. Furthermore, the summarized AI values of the above six odor classes for different WRTs were presented as radar graphs (Figure 4), and four similar aroma profiles were observed with the predominant performances in “floral & fruity” (42.0−49.1 of total AI values) and “green & fresh” attributes (43.2−46.0 of total AI values), moderate performance in “roasted and caramel” (21.8−26.4 of total AI values), and nearly equal and lower proportions in other attributes (9.6−16.8 of total AI values). Specifically, the AIs of the “floral & fruity” attribute of RG and JF were obviously higher than those of DHP and SX, while the opposite appearance was observed for the “roasted and caramel” attribute. The highest AI values of “floral & fruity” (49.1) and “green & fresh” (46.0) were exhibited by the RG sample. Additionally, the lowest AI value of the “green & fresh” attribute was obtained in JF (43.2), while its “sweet” attribute (16.8) was significantly higher than in other types of WRTs.

For details, benzeneacetaldehyde (3.4, class A), methyl salicylate (3.3, class B), 2-methyl-3,5-diethylpyrazine (3.3, class C), acetophenone (3.4, class D) in DHP; phenethyl acetate (3.4, class A), *trans*-alloocimene (3.3, class B), acetophenone (3.2, class D), and (*E*,*E*)-2,4-heptadienal (3.3, class F) in RG; (*E*,*Z*)-2,6-nonadienal (3.3, class B), 2-methyl-3,5-diethylpyrazine (3.3, class C), acetophenone (3.2, class D), (*E*,*E*)-2,4-heptadienal (3.7, class F) in SX; *trans*-β-ocimene (3.3, class A), linalool (3.3, class A), *trans*-alloocimene (3.4, class B), 2-acetylpyrrole (3.2, class C), and acetophenone (3.2, class D) in JF presented superior (>3.0) and the highest intensities in the corresponding WRT samples.

By comparing the intensities of the aroma-active compounds in WRTs with different cultivars, methyl octanoate (3.0−3.2), *trans*-linalool oxide (furanoid) (3.0−3.2), benzeneacetaldehyde (3.0−3.4), *trans*-β-ocimene (3.0−3.3), *trans*-alloocimene (3.2−3.4), 2-acetylpyrrole (3.0−3.2), and acetophenone (3.2−3.4) generally had strong aroma intensities (AI ≥ 3.0), but there were no significant differences among the samples, which may be due to the common and basic aroma-active components of WRTs. The above compounds also play important roles in the aroma formation of most kinds of teas, especially in black and oolong teas, which frequently present a typical floral scent [1,31]. Notably, some odorants, such as indole (1.0−2.5), methyl salicylate (2.4−3.3), 2,5-dimethylpyrazine (2.4−3.2), and *p*-cymene (2.0−3.2), had obvious gaps in the AI values among different WRTs. Moreover, some odorants could only be detected in at most two types of WRTs, e.g., 2-methyl-3,5-diethylpyrazine (nutty) could be detected in the SX and DHP samples, coumarin (milk-like, sweet, floral) could be detected in the other two WRTs, and 2,2,6-trimethyl-cyclohexanone (sweet, cooked rice) was only detected in the JF sample. These compounds may be potential odorants that reflect the aroma characteristics of WRTs from different cultivars. 2-Methyl-3,5-diethylpyrazine has been proven to be the key odorant responsible for the special roasted and caramel-like aroma of DHP tea [17]. Coumarin has been reported as the odorant of Jingshan cha and Longjing green teas [32,33], while it was firstly recognized as the aroma-active compound in oolong teas. However, because of the inevitable sensory errors caused by olfactory sensitivity differences and subjective influences from different panelists, a comprehensive analysis of the GC-O/MS and stoichiometry results was necessary, and the compounds recognized as both key differential VFCs and aroma-active compounds were further analyzed.

### 3.4. Determination of Characteristic Volatiles in WRTs with Different Cultivars

A total of 18 key differential VFCs were recognized as the aroma-active compounds in WRTs, most of which belonged to class A and class C with the “floral & fruity” and “roasted & caramel” scents in Table 1, and they were defined as the potential characteristic key odorants in the WRTs with different cultivars. In theory, the concentration distribution trends of the potential key odorants were supposed to be proportional to the order of their AI values among different WRTs. However, some inconsistent appearances between the stoichiometry and sensory results were also observed, probably due to instrumental and sensory errors, as well as high odor thresholds but not sufficiently large concentration differences among different WRTs presented by some odorants. To ensure the accuracy of the characteristic key odorants in WRTs with different cultivars, the potential odorants with inconsistent distribution trends between the average peak areas and AI values were removed. Sixteen odorants were maintained, with various distribution trends among the four types of WRTs (Figure 5). Methyl salicylate, *p*-cymene, 2,5-dimethylpyrazine, and 1-furfurylpyrrole presented the highest concentration levels (*p* < 0.05) and AIs (2.8–3.3) in the DHP samples; thus, they were identified as the characteristic key odorants of DHP. The pyrrole and pyrazine derivatives were believed to be newly formed through Maillard reactions during WRT processing, especially after the final full firing step [3,12], which were generated from Strecker degradations between theanine and *D*-glucose under high temperatures [4,9]. Although 2,5-dimethylpyrazine and 1-furfurylpyrrole have been reported as flavor components or odorants in SX [3,14] and RG [5,7] samples, they have never been identified as unique odorants in the corresponding samples, which is consistent with our results on the secondary side. Methyl salicylate and *p*-cymene are common aromatic VFCs in teas, which are derived from the biosynthesis of the shikimate pathway in tea plants [34,35], and the free form of methyl salicylate is released via the hydrolysis of their glycosidically bound forms under enzymatic or thermal reactions during tea processing [36]. In addition, methyl salicylate has been reported to be one of the odorants in DHP, but its contribution to the overall aroma is far less than that in this study [17]. The differences in extraction and detection methods may be the main reasons for the differences.

Similarly, phenethyl acetate, phenylethyl alcohol, indole, and (*E*)-β-famesene were identified as the characteristic key odorants of RG, both of which presented “floral” or “fruity” scents. Among them, phenylethyl alcohol was proven to significantly influence the aroma perception of RG [5]; it was not detected in fresh tea leaves, but exhibited a growing trend in the concentrations with the promotion of RG processing, and its glycoside precursors may be heavily hydrolyzed after fermentation [6]. In addition, phenethyl acetate, indole, and (*E*)-β-famesene have been reported as the characteristic aroma compounds of oolong tea [13,37], and their precursors are speculated to be phenylalanine, tryptophan, and carotenoids [38].

Linalool, phenethyl butyrate, hexyl hexanoate, and dihydroactinidiolide were identified as the key odorants of JF, with the highest concentrations and AI levels. The first three odorants were reported to exist in the tea leaves, indicating that the characteristics might have originated from their own property from cultivar differences, while dihydroactinidiolide, sourced from the complex oxidation of β-carotene, was mainly generated during tea processing [4], and it was reported to be detected after roasting during RG manufacturing [6].

Compared with the other WRTs, the concentrations of key differential VFCs in the SX samples were generally unattractive, and thus, the number of characteristic key odorants of SX was fewer than that of other WRTs. Only naphthalene presented the highest average concentrations and AI values in SX, which has been identified as a common VFC among WRTs with SX, Huangmeigui, and Zimudan cultivars [14].

In addition, some potential odorants presented significantly high but equal concentrations and AI value levels in both types of WRTs, including *N*-ethylpyrrole and 2-acetylfuran in the DHP and SX samples and *trans*-linalool oxide (furanoid) in the RG and JF samples, and they were considered as the common key odorants to distinguish between the two cultivars. *N*-Ethylpyrrole and 2-acetylfuran with roasted and baked scents were similarly generated from theanine and *D*-glucose during the roasting steps of the WRTs, as mentioned above. *trans*-Linalool oxide (furanoid) naturally exists as free or glycosidically bound forms in fresh tea leaves, and it was reported to sharply increase after the roasting steps of RG, which might be due to the pyrolysis of its glycoside precursors at high temperatures [6]. Moreover, the odor characteristics of the above compounds were highly consistent with the preponderant aroma attributes of DHP and SX (roasted) and RG and JF (floral) shown in Figure 4, and thus the odorants might contribute much to the formation of the sensory differences.

## 4. Conclusions

In summary, a comprehensive characterization of key differential VFCs and odorants in WRTs from four different cultivars was performed using GC-O/MS, chemometrics, and multivariate statistical analysis. Sixteen aroma-active compounds were identified as the characteristic key odorants among the four types of WRTs, especially methyl salicylate, *p*-cymene, 2,5-dimethylpyrazine, and 1-furfurylpyrrole in DHP; phenylethyl alcohol, phenethyl acetate, indole, and (*E*)-β-famesene in RG; linalool, phenethyl butyrate, hexyl hexanoate, and dihydroactinidiolide in JF; and naphthalene in SX. The obtained results have enriched the theoretical knowledge of tea biochemistry and provide a fundamental basis for distinguishing tea cultivars, recombination, and simulation of the WRT aroma.

## Figures and Tables

**Figure 1 foods-11-04109-f001:**
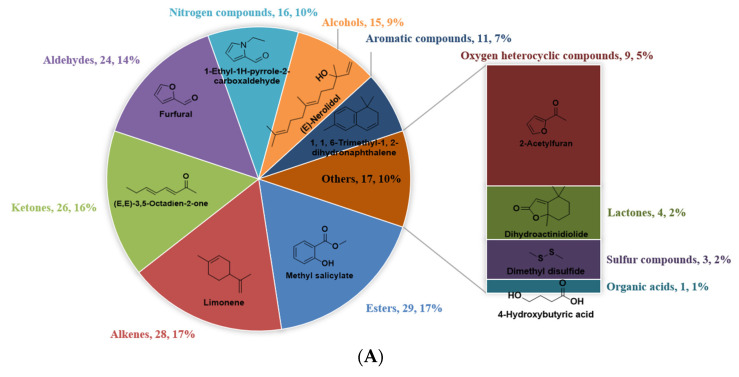
Number distribution and the compounds with top concentrations on the 11 chemical categories of volatile fragrant compounds (VFCs) in Dahongpao (DHP; (**A**)), Rougui (RG; (**B**)), Shuixian (SX; (**C**)) and Jinfo (JF; (**D**)) samples.

**Figure 2 foods-11-04109-f002:**
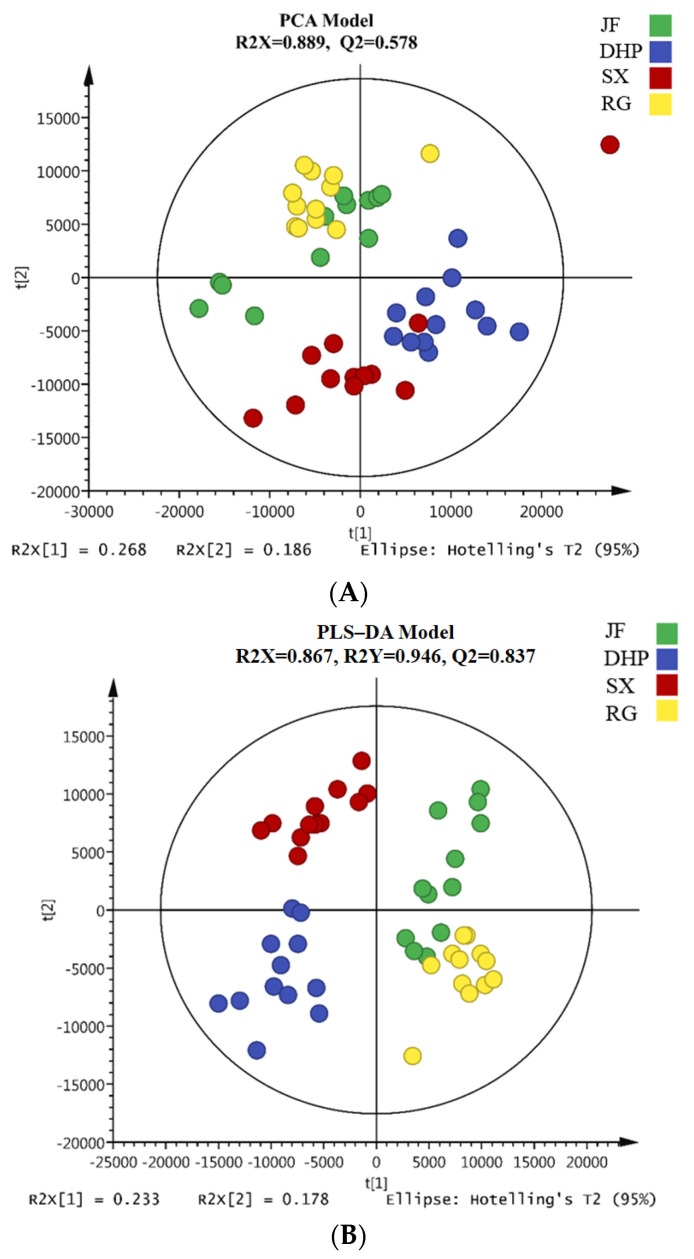
(**A**) Principal component analysis (PCA) model of volatile fragrant compounds (VFCs) in Wuyi rock teas (WRTs) with four different cultivars; (**B**) Partial least squares discriminant analysis (PLS–DA) model of VFCs in WRTs with four different cultivars; (**C**) Validation model of the PLS–DA model.

**Figure 3 foods-11-04109-f003:**
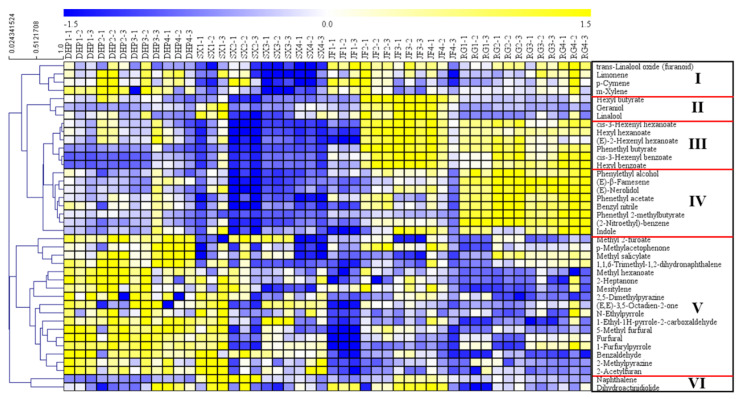
Heatmap based on the normalized peak area of the key differential volatile fragrant compounds (VFCs) among Wuyi rock teas (WRTs) with four different cultivars.

**Figure 4 foods-11-04109-f004:**
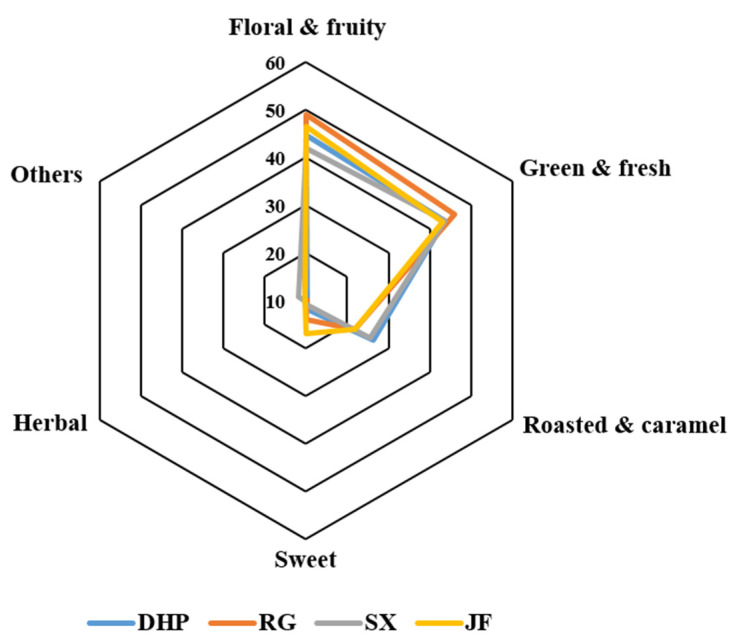
Radar graph based on the total aroma intensity (AI) values of the six odor classes for the four types of Wuyi rock teas (WRTs).

**Figure 5 foods-11-04109-f005:**
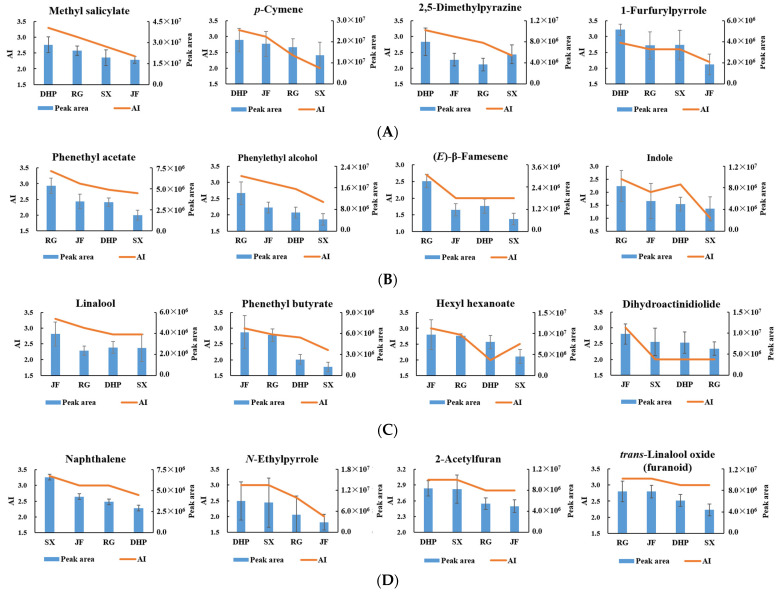
Characteristic key odorants of different types of Wuyi rock teas (WRTs) ((**A**): Dahongpao (DHP); (**B**): Rougui (RG); (**C**): Jinfo (JF); (**D**): Shuixian (SX), DHP and SX, JF and RG).

**Table 1 foods-11-04109-t001:** Odor characteristics and aroma intensities (AIs) of the aroma-active compounds in Wuyi rock teas (WRTs) with different cultivars.

No	Compound ^a^	Odor Characteristic ^b^	AI				Identification ^c^
			DHP	RG	SX	JF	
Class A (floral & fruity)							
1	Benzeneacetaldehyde	Rose-like	3.4	3.0	3.0	3.0	MS, RI, O, S
2	Methyl octanoate	Fruity, sweet, green	3.2	3.0	3.0	3.0	MS, RI, O
3	*trans*-Linalool oxide (furanoid) *	Floral, sweet	3.0	3.2	3.0	3.2	MS, RI, O, S
4	*trans*-β-Ocimene	Floral, herbal	3.0	3.0	3.0	3.3	MS, RI, O, S
5	Linalool *	Floral, sweet	2.8	3.0	2.8	3.3	MS, RI, O, S
6	Phenylethyl alcohol *	Floral, sweet	2.8	3.2	2.4	3.0	MS, RI, O, S
7	Phenethyl acetate *	Floral, cucumber, sweet	2.8	3.4	2.7	3.0	MS, RI, O
8	*trans*-β-Ionone	Rose-like, coconut creamy	2.8	2.8	3.0	3.0	MS, RI, O, S
9	Phenethyl butyrate *	Floral, fruity, sweet	2.7	2.8	2.3	3.0	MS, RI, O
10	Methyl benzoate *	Floral, leather	2.5	2.8	3.0	2.7	MS, RI, O, S
11	(*E*)-2-Hexenyl hexanoate *	Fruity, floral, sweet	2.5	2.8	2.5	3.0	MS, RI, O, S
12	α-Ionone	Floral, sweet	2.5	2.3	1.5	1.5	MS, RI, O, S
13	Indole *	Floral, leather	2.3	2.5	1.0	2.0	MS, RI, O, S
14	α-Cedrene	Floral, sweet, fruity	2.3	2.8	2.3	2.4	MS, RI, O, S
15	Octanal	Fruity, green, fatty	2.0	3.0	2.0	2.0	MS, RI, O, S
16	Hexyl hexanoate *	Fruity, floral	2.0	2.8	2.5	3.0	MS, RI, O, S
17	(*E*)-β-Famesene *	Floral, sweet	2.0	2.7	2.0	2.0	MS, RI, O, S
Class B (green & fresh)							
1	Methyl salicylate *	Minty, wintergreen-like, herbal	3.3	3.0	2.7	2.4	MS, RI, O, S
2	*trans*-Alloocimene	Green, fresh, floral	3.2	3.3	3.2	3.4	MS, RI, O
3	(*E*,*E*)-2,4-Nonadienal	Green, fruity	3.2	3.2	2.8	2.8	MS, RI, O, S
4	(*E*,*Z*)-2,6-Nonadienal	Green, cucumber	3.2	2.8	3.3	3.3	MS, RI, O
5	(*E*)-2-Octenal	Green, fresh, nut	3.0	3.0	3.2	2.6	MS, RI, O, S
6	Nonanal	Floral, fresh	2.8	3.0	2.8	3.3	MS, RI, O, S
7	Benzyl acetate	Fresh, green	2.8	2.8	2.8	2.8	MS, RI, O, S
8	Propiophenone	Green, fresh	2.8	2.8	2.8	2.8	MS, RI, O
9	Methylheptenone	Green, lemon, floral	2.8	2.7	2.7	3.2	MS, RI, O, S
10	(*E*)-2-Nonenal	Green, cucumber	2.6	3.0	3.0	2.5	MS, RI, O, S
11	5-Ethyl-6-methyl-3E-hepten-2-one	Green, fresh	2.6	2.5	2.7	2.5	MS, RI, O
12	1-Octen-3-ol	Mushroom-like, fresh	2.5	3.0	2.7	2.6	MS, RI, O, S
13	(*E*)-2-Decenal	Green, fresh	2.5	2.8	3.0	2.5	MS, RI, O
14	Hexanal	Green, grassy	2.3	2.8	1.8	2.2	MS, RI, O, S
15	α-Citral	Citrus, fresh	2.0	2.3	2.0	2.3	MS, RI, O
16	Calamenene	Fresh and cool	2.0	3.0	2.0	2.0	MS, RI, O, S
Class C (roasted and caramel)							
1	2-Methyl-3,5-diethylpyrazine	Nutty	3.3	-	3.3	-	MS, RI, O
2	2,5-Dimethylpyrazine *	Barked, coffee	3.2	2.8	2.4	3.0	MS, RI, O, S
3	2-Acetylpyrrole	Caramel	3.0	3.0	3.0	3.2	MS, RI, O, S
4	2-Acetylfuran *	Roasted, cocoa	3.0	2.8	3.0	2.8	MS, RI, O, S
5	*N*-Ethylpyrrole *	Burnt, barked	3.0	2.6	3.0	2.0	MS, RI, O, S
6	3-Ethyl-2,5-dimethylpyrazine	Roasted, coffee	3.0	2.4	3.2	2.8	MS, RI, O
7	2,2’-Methylenebis-furan	Roasted, bitter	2.8	2.8	2.4	3.0	MS, RI, O
8	1-Furfurylpyrrole *	Roasted, green	2.8	2.6	2.6	2.2	MS, RI, O
9	2-Methylpyrazine	Roasted, pesticide-like	2.3	2.8	2.5	2.8	MS, RI, O, S
Class D (sweet)							
1	Acetophenone	Sweet, floral	3.4	3.2	3.2	3.2	MS, RI, O
2	*p*-Cymene *	Fragrant and sweet, fresh, rice-like	3.2	2.4	2.0	3.0	MS, RI, O
3	6-Methyl-3,5-heptadiene-2-one	Sweet, coconut creamy	3.0	3.0	3.0	2.3	MS, RI, O, S
4	γ-Nonanolactone	Coconut creamy, sweet, floral	2.0	2.5	2.5	2.5	MS, RI, O, S
5	Coumarin	Milk-like, sweet, floral	-	2.8	-	3.0	MS, RI, O, S
6	2,2,6-Trimethyl-cyclohexanone	Sweet, cooked rice	-	-	-	2.8	MS, RI, O, S
Class E (herbal)							
1	3,5-Octadien-2-one	Mushroom-like	3.0	2.8	3.0	2.8	MS, RI, O, S
2	3-Octen-2-one	Herbal, fruity	2.8	2.8	3.0	2.8	MS, RI, O
3	Safranal	Herbal, metallic	2.5	2.0	2.3	1.7	MS, RI, O
4	Dihydroactinidiolide *	Herbal, essential oil	2.0	2.0	2.0	3.0	MS, RI, O, S
Class F (others)							
1	Heptanal	Fatty, green, herbal	2.8	2.0	2.6	2.8	MS, RI, O, S
2	Naphthalene *	Pungent, earthy	2.7	3.0	3.3	3.0	MS, RI, O
3	Dimethyl trisulfide	Sulfurous, pickled vegetables	2.3	2.0	2.3	2.2	MS, RI, O, S
4	(*E*,*E*)-2,4-Heptadienal	Fatty, slight pungent, green	2.0	3.3	3.7	2.3	MS, RI, O, S

^a^: The compounds with asterisks (*) were also identified as the key differential volatile fragrant compounds (VFCs) among Wuyi rock teas (WRTs) with different cultivars; ^b^: the odor characteristics of the aroma-active compounds were summarized from the comments provided by the six panelists; ^c^: the structures of aroma-active compounds were identified by the comparison of mass spectra (MS), retention index (RI), http://www.thegoodscentscompany.com/search2.html (accessed on 9 March 2022) (O) and commercial standards (S).

## Data Availability

Data will be made available on request.

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
