# Peer review of "Insights into Characteristic Volatiles in Wuyi Rock Teas with Different Cultivars by Chemometrics and Gas Chromatography Olfactometry/Mass Spectrometry"

_foods, 2022, doi:10.3390/foods11244109_

Round 1

Reviewer 1 Report

Dear Authors,

In general, the work described in the manuscript is quite innovative and of practical significance to the development of modern industry. The manuscript is well organized and the result are reasonable.

3.1, 3.2, 3.3 There is no discussion nor references in all these three sections of Results and Discussion.

The authors are invited to add a substantial discussion in order to justify and support the findings of the study.

There are novel and recent papers on this topic that could be mentioned here.

Author Response

Point 1: 3.1, 3.2, 3.3 There is no discussion nor references in all these three sections of Results and Discussion. The authors are invited to add a substantial discussion in order to justify and support the findings of the study. There are novel and recent papers on this topic that could be mentioned here.

Point 1: In general, the work described in the manuscript is quite innovative and of practical significance to the development of modern industry. The manuscript is well organized and the result are reasonable.

3.1, 3.2, 3.3 There is no discussion nor references in all these three sections of Results and Discussion. The authors are invited to add a substantial discussion in order to justify and support the findings of the study. There are novel and recent papers on this topic that could be mentioned here.

Response 1: According to the reviewer 1’s suggestion, a substantial discussion based on the results obtained in our study and those reported in previous references was added in the 3.1, 3.2, 3.3 sections of revised manuscript. See detailed revisions on lines 186-192, 200-210 of pages 4-5; lines 291-293, 301-303, 313-320 of pages 10-11; lines 334-337, 380-382, 389-393 of pages 12 and 14. Besides, the mentioned literatures were added in the section of Reference.

We acknowledge the reviewer for his/her cautious and conscientious work!

Reviewer 2 Report

This manuscript reports the flavor differences of Chinese tea between several cultivars. As the analytical methodology, not only GC-MS, but also GC-O was applied to evaluate flavor differences. Those approach look novel. However, data interpretation seems redundant and several revisions are required.

Whole manuscript:

The word “identification” should not be used in this manuscript. To use the word “identify”, retention time matching with authentic standard using two different stationary phase of GC columns is required (eg. Molyneux et al. (2007) J. Agr. Food Chem., 55, 4625-4629). Since the method of chemical analysis had used only one column (BD-5MS), this manuscript can use the word “tentatively identified”.

Introduction:

The reason why those tea flavor should be distinguished is needed as the purpose of this research.

Figure 1:

Too small to read.

Figure 1 and Figure 3:

Those two figures indicate almost same results and discussion seems redundant. One of those might be deleted.

Figure 4:

This figure dose not make sense in this section. The discussion of this figure is same as Table 1. One of those might be deleted.

Figure 5:

Too small to read. Chose a few representative chemicals and show them larger. It needs 1st and 2nd legend of Y axis.

Author Response

Point 1: This manuscript reports the flavor differences of Chinese tea between several cultivars. As the analytical methodology, not only GC-MS, but also GC-O was applied to evaluate flavor differences. Those approach look novel. However, data interpretation seems redundant and several revisions are required.

Whole manuscript: The word “identification” should not be used in this manuscript. To use the word “identify”, retention time matching with authentic standard using two different stationary phase of GC columns is required (eg. Molyneux et al. (2007) J. Agr. Food Chem., 55, 4625-4629). Since the method of chemical analysis had used only one column (BD-5MS), this manuscript can use the word “tentatively identified”.

Response 1: According to the reviewer’s suggestion, “identification” has been modified to “tentatively identified” (line 183 of page 4), and the related discussion has been added in the revised manuscript (lines 188-192 of page 4).

Point 2: Introduction: The reason why those tea flavor should be distinguished is needed as the purpose of this research.

Response 2: According to the reviewer’s suggestion, the reason for the discrimination of different WRTs was added in the Section of Introduction (lines 54-58 of page 2).

Point 3: Figure 1: Too small to read. 

Response 3: According to the reviewer’s suggestion, the original Figure 1 has been divided into four parts (Figure 1A-D) to clearly display the corresponding contents in the revised manuscript.

Point 4: Figure 1 and Figure 3: Those two figures indicate almost same results and discussion seems redundant. One of those might be deleted.

Response 4: Probably due to the unclear display in the original Figure 1, misunderstanding of the Figure 1 and Figure 3 were occurred. In fact, the contents of Figure 1 were 1) number distribution of the identified VFCs with different chemical categories in WRTs; 2) the compounds with top concentrations in each group of chemical categories; and the results indicated the similar and common distribution among WRTs with different cultivars. Inversely, the compounds in Figure 3 were the key differential VFCs among different WRTs, which was the further anlaysis results distinguished from Figure 1. Therefore, significant differences on the contents between the two Figures were presented, and the two Figures were maintained in the revised manuscript, and a more clear version of Figure 1 was replaced. 

Point 5: Figure 4: This figure dose not make sense in this section. The discussion of this figure is same as Table 1. One of those might be deleted.

Response 5: The Figure 4 was a summary of the aroma profile of differernt WRTs based on the total AI values of the six odor classes. Although the separate AI value of each aroma-active compound was showed in Table 1, the overall aroma profiles of WRTs were unable to be displayed without Figure 4. The contents of Figure 4 indicated that four similar aroma profiles were observed with the predominant performances in “floral & fruity” (42.0−49.1 of total AI values) and “green & fresh” attributes (43.2−46.0 of total AI values), moderate performance in “roasted and caramel” (21.8−26.4 of total AI values), and nearly equal and lower proportions in other attributes (9.6−16.8 of total AI values). In order to better present Figure 4, the redundant part of the original Figure 4 has been deleted, and more intuitive version was replaced.

Point 6: Figure 5: Too small to read. Chose a few representative chemicals and show them larger. It needs 1st and 2nd legend of Y axis.

Response 6: According to the reviewer’s suggestion, the original Figure 5 has been divided into four parts (Figure 5A-D) to clearly display the corresponding contents in the revised manuscript, and 1st and 2nd legend of Y axis for each odorant has been added. In fact, at most 4 compounds for each type of WRTs were presented in the Figure 5, and all of which were the representative odorants. Thus, the compounds in original Figure 5 was maintained.

We acknowledge the two reviewers for his/her cautious and conscientious work!

Round 2

Reviewer 1 Report

Dear Authors,

Thank you for the new version of your manuscript.

My comments on it are the following:

The titles of Tables and Figures should describe in full all acronyms. Please add the full name of all treatments.

Author Response

Point 1:  The titles of Tables and Figures should describe in full all acronyms. Please add the full name of all treatments.

Response 1: According to the reviewer 1’s suggestion, the full names of all acronyms in Tables and Figures of manuscript and Supplementary Material have been added.

We acknowledge the reviewer for his/her cautious and conscientious work!
